

# The fight against malaria in Edo-North, Edo State, Nigeria: identifying risk factors for effective control

Joseph Odunayo Braimah[1,2], Nnamdi Edike[2], Augustine Ijeameran Okhaiomoje[3] and Fabio Mathias Correa[1]

[1] Department of Mathematical Statistics and Actuarial Sciences, University of the Free State, Bloemfontein, Free State, South Africa
[2] Department of Mathematics and Statistics, Ambrose Alli University, Ekpoma, Edo State, Nigeria
[3] Department of Laboratory Sciences, Ambrose Alli University, Ekpoma, Edo State, Nigeria

Corresponding author
Joseph Odunayo Braimah,
braimahjosephodunayo@
aauekpoma.edu.ng

## ABSTRACT

**Background:** This study investigated malaria epidemiology in Edo-North, Nigeria; a region within the equatorial rainforest belt that has lacked prior research on malaria prevalence. This research sought to investigate the prevalence of malaria and identify potential risk factors in Edo-North, Nigeria. Additionally, the study aimed to analyze trends in malaria cases to inform the development of effective malaria control measures.

**Methodology:** A cross-sectional study was conducted in six local government areas of Edo-North, Nigeria, between June and August 2023. Using systematic sampling, study zones, local governments, towns, villages, and households were selected. Data on sociodemographics and environmental risk factors were collected from 605 participants through questionnaires and blood samples. Blood smears were microscopically examined, binary and multivariate logistic regression was used for data analysis. Malaria disease rate trends were also analyzed from health records. Statistical analyses were performed using R software, with $p$-values less than 0.05 considered statistically significant.

**Results:** The overall malaria prevalence in the study area was 15.54%, with males more likely to be infected than females. Prevalence varied across localities, with Akoko-Edo having the highest rate. Children had the highest prevalence. Rural residents were more likely to have malaria than urban residents. Binary logistic regression identified several risk factors, including age, location, local government area, education, occupation, marital status, housing type, household size, water source, sanitation, surrounding environment, window net use, ceiling type, water storage, and parasite density. The multivariate logistic regression analysis identified several significant risk factors for malaria in the study population. Children, residents of Owan-East, individuals using pit latrines, and those not sleeping under LLINs were at significantly increased risk. Environmental factors such as proximity to bushes, streams/rivers, and storing water in open containers were also associated with higher malaria prevalence. History of malaria treatment at pharmacies and use of Chloroquine/Quinine medication were linked to recurrent infections. The study found a high average parasite density (5,146 parasites/μL) and low consistent LLIN use despite widespread ownership. Trend analysis from malaria records revealed a decline in malaria prevalence from 2020 to 2023.

**Conclusions:** The study identified several demographic, environmental and behavioural factors associated with malaria risk in Edo North. Males, children, urban dwellers, those living in mud houses and those in large households, proximity to natural features such as bushes, rivers and streams and low LLIN were more likely to contract malaria. These findings highlight the importance of implementing targeted interventions to address these risk factors and reduce the burden of malaria.

# INTRODUCTION

Malaria, caused by Plasmodium parasites is a potentially fatal disease that humans contract by the bite of an infected female Anopheles mosquito. Malaria is still a serious global public health issue. Though it is preventable and curable; nearly 95% of cases and fatalities occur in Nigeria, which is the country with the highest economic burden from the disease. Four major Plasmodium species (*Plasmodium malariae*, *Plasmodium falciparum*, *Plasmodium vivax*, and *Plasmodium ovale*) are responsible for almost 200 million cases reported each year in African countries (*World Health Organization (WHO), 2019*). Malaria requires substantial personal and governmental financial resources for prevention and treatment, in addition to its high rates of morbidity and mortality throughout the African continent (*World Health Organization (WHO), 2017*; *Ogbuabor et al., 2023*). It consequently has an impact on the wealth and health of the nation and is a primary contributor to poverty in many African nations (*Ogbuabor et al., 2023*; *Abubakar et al., 2022*). The Global Technical Strategy (GTS) for Malaria 2016–2030, a global drive to end malaria by 2030 with precise targets and milestones, was created by the World Health Organization (WHO) in response to this threat (*World Health Organization (WHO), 2024*). The WHO recommends that nations adopt context-specific interventions that prioritize early diagnosis, timely treatment, vector control, and elimination of vectors in accordance with this strict strategy (*World Health Organization (WHO), 2019*, *2024*).

Many African nations are far from reaching the GTS target, regardless of their best efforts (*Apinjoh et al., 2015*; *Nyasa et al., 2015*; *World Health Organization (WHO), 2023*). The fact that adults exposed to Plasmodium species constitute a serious threat to attempts to control and eradicate malaria, they are frequently disregarded in these nations in favor of pregnant women and children under five. In order to effectively accomplish WHO's objectives, new approaches must be devised to reach all populations, including adults infected with *Plasmodium* spp., and existing programs must be enhanced. In highly endemic environments like Sub-Saharan Africa, adult Plasmodium species infections constitute a primarily composite challenge for the pathogenicity and epidemiology of malaria (*Kwenti et al., 2017*). Actually, more exposure to this species of Plasmodium has resulted in the development of anti-parasitic immunity, which has also reduced blood parasite levels, prevented symptoms, and offered notable protection against acute malaria and mortality associated with it (*Hay, Smith & Snow, 2008*; *Teh et al., 2019*). Adults
naturally accumulate more infections over time, and the majority of them acquire immunity to the malaria parasite, unless they have an underlying medical condition like pregnancy or HIV/AIDS (*Hay, Smith & Snow, 2008*).

Acquired immunity, however, is still only partially effective at preventing infections. Adults who are semi-immune can still have Plasmodium species infections, although exhibiting few or no symptoms and having the parasites in their blood for extended periods of time (*Teh et al., 2019*; *Sumbele, Nkemnji & Kimbi, 2017*; *Teh et al., 2018*). Community surveys conducted in sub-Saharan Africa reveal Plasmodium species prevalence estimates that are both over and below 5% amongst asymptomatic persons (*Essangui et al., 2019*; *Tougan et al., 2020*; *Ibrahim et al., 2023*; *Ngum et al., 2023*; *Dawaki et al., 2016*; *Nmadu et al., 2015*). Different current vector species, different intervention coverage, and broad ecological changes (such as altitude temperature, and seasons) are likely to be associated with regional variations. Long-term consequences, including anemia, cognitive decline, and recurring symptomatic episodes, may arise from persistent infections with Plasmodium spp., despite the fact that these infections are frequently asymptomatic (*Sumbele, Nkemnji & Kimbi, 2017*; *Tougan et al., 2020*). In the end, they would significantly raise healthcare expenses, decrease productivity, and put more financial strain on African communities already dealing with other social problems. Furthermore, typical diagnostic methods, such as conventional microscopy and rapid diagnostic tests (RDTs), are greatly hampered by the fact that infected adults in endemic areas frequently have low blood levels of parasites (*World Health Organization (WHO), 2023*; *Sumbele, Nkemnji & Kimbi, 2017*; *Teh et al., 2018*).

## Malaria in Nigeria

Variations in socio-demographic, environmental, and climatic factors may account for the variation in the prevalence rate of malaria infection among young people, even within a single country (*Ibrahim et al., 2023*; *Ngum et al., 2023*). Past research has shown malaria prevalence rates in Nigeria of 66.7% (*Dawaki et al., 2016*), 64.0% (*Nmadu et al., 2015*), and 58.0% (*Awosolu et al., 2021*). Urban and suburban centers have comprehensively recorded the effects of socio-demographic variables or factors, including gender, age, occupation, and level of education, on human exposure and treatment, as well as environmental variables like rainfall, humidity, and temperature, may facilitate the rapid growth of mosquitos' vectors (*Ibrahim et al., 2023*; *Awosolu et al., 2021*). The spread of malaria is more common in Africa's rural settings than in its urban ones, which may be caused by the region's increased vector density, subpar housing conditions, and inadequate drainage infrastructure (*Ngum et al., 2023*; *Dawaki et al., 2016*; *Nmadu et al., 2015*; *Awosolu et al., 2021*; *Oladeinde et al., 2012*). An increasing amount of research indicates that children and teenagers attending school may potentially be susceptible to malaria (*World Health Organization (WHO), 2023*; *Squire et al., 2016*). Furthermore, the prime age of clinical bouts of malaria is moving from young infants and adolescents to children and teenagers due to a drop in spread and exposure in some locations (*World Health Organization (WHO), 2023*; *Squire et al., 2016*). Data on P. falciparum infection prevalence and related variables among teenage age groups in rural Nigeria are desperately needed to be able to

modify the prevention and control of malaria strategies. For the purpose of malaria prevention and control strategies, decision-makers must identify the major risk factors for malaria infection.

Currently, the most common approaches for managing malaria vectors and related malaria transmission are indoor residual spraying (IRS) and insecticide-treated nets (ITNs), both of which have been shown to decrease malaria (*Abubakar et al., 2022*; *World Health Organization (WHO), 2024*). Nonetheless, the primary and most encouraging element of the specific vector control and management practice are long-lasting insecticide-treated nets (LLINs) on sleeping beds, which the Nigerian Ministry of Health uses to combat the disease and its vector.

## Nigeria's malaria control strategy

Nigeria's fight against malaria has made significant progress, underpinned by a robust control strategy. Key components include surveillance, bed net distribution and health promotion.

## Surveillance

Nigeria has a robust surveillance system to monitor the malaria burden and inform interventions. The National Malaria Indicator Survey (NMIS) provides key data on malaria prevalence, risk factors and intervention coverage. This surveillance system is central to identifying hotspots, tracking trends and evaluating the impact of control interventions (*Addo et al., 2021*; *Amede et al., 2022*).

## Net distribution strategy

The distribution of long-lasting insecticidal nets (LLINs) is a cornerstone of malaria control in Nigeria. Mass distribution campaigns have been instrumental in increasing LLIN ownership. The National Malaria Elimination Programme (NMEP) oversees these campaigns, which target vulnerable populations, particularly pregnant women and children under (*Federal Ministry of Health, 2022*; *World Health Organization (WHO), 2022*, *2024*).

## Health promotion campaigns

Nigeria has implemented extensive health promotion campaigns to raise awareness of malaria prevention and treatment. Messages focus on early diagnosis, prompt treatment with artemisinin-based combination therapies (ACTs) and consistent use of LLINs. Community-based initiatives involving traditional and religious leaders have been effective in disseminating information and changing behavior (*Akpan et al., 2023*; *Mehta, 2024*).

## Challenges and future directions

Despite progress, Nigeria faces challenges in achieving malaria elimination, including inadequate funding, resistance to antimalarial drugs, and gaps in access to healthcare. Future efforts should focus on strengthening surveillance, expanding access to diagnosis and treatment, and developing innovative strategies to address emerging challenges (*Jeremiah et al., 2023*).

Therefore, the goal of this study was to ascertain the malaria risk factors and trend in the study populations.

## MATERIALS AND METHODS

### Study location

The study took place in four local government areas of Edo-North, a region located within Edo state, Nigeria. Edo-North includes eight of the state's eighteen local governments, specifically Akoko-Edo, Etsako-West, Etsako-Central, Etsako-East, Owan-East, and Owan-West (Fig. 1). Edo state is one of the states located in the rain forest zone in Nigeria, with an average annual rainfall between 1,800 and 2,780 millimeters and a mean temperature of 28 degrees Celsius. Malaria continues to be a significant health problem in the region, but transmission intensity varies among the different local governments. Edo North District was chosen for this malaria epidemiology study due to its high malaria burden, lack of previous data, predominantly rural population, and diverse ecological setting. The region's lowlands, sloping streams, and rainforest vegetation may impact the breeding and distribution of mosquito vectors.

### Sampling technique

A systematic sampling approach was employed to select six local government areas in Edo North. Within each selected local government, towns and villages were systematically chosen to ensure representative coverage. Households were then randomly selected from these locations. Data on sociodemographics and environmental risk factors were gathered from 605 participants using questionnaires and blood samples.

### Study design

There were two parts to the study. A cross-sectional survey was carried out during August and September 2023 to assess the prevalence of malaria infections and associated risk factors within households, a cross-sectional household survey conducted in the selected communities that took place in September 2023, when most people had returned from their farms, and involved meeting participants at their homes in cool evening to conduct personal interviews. Blood samples were also taken for microscopy and sent for testing at the Irrua Specialists teaching hospital in Irrua, Edo State. In order to determine trends in the prevalence of malaria in Edo-North, data on hospital prevalence from 2021 to 2023 were collected retrospectively from Edo-North's Public Health Centers as part of the study's second arm.

### Exclusion criteria for the study

Individuals who declined to provide their consent, those who did not reside in Edo-North, those suffering from mental health issues, and those receiving anti-malarial treatment or those who has received anti-malarial treatment 2 weeks before the commencement of this study were also not included in the study.

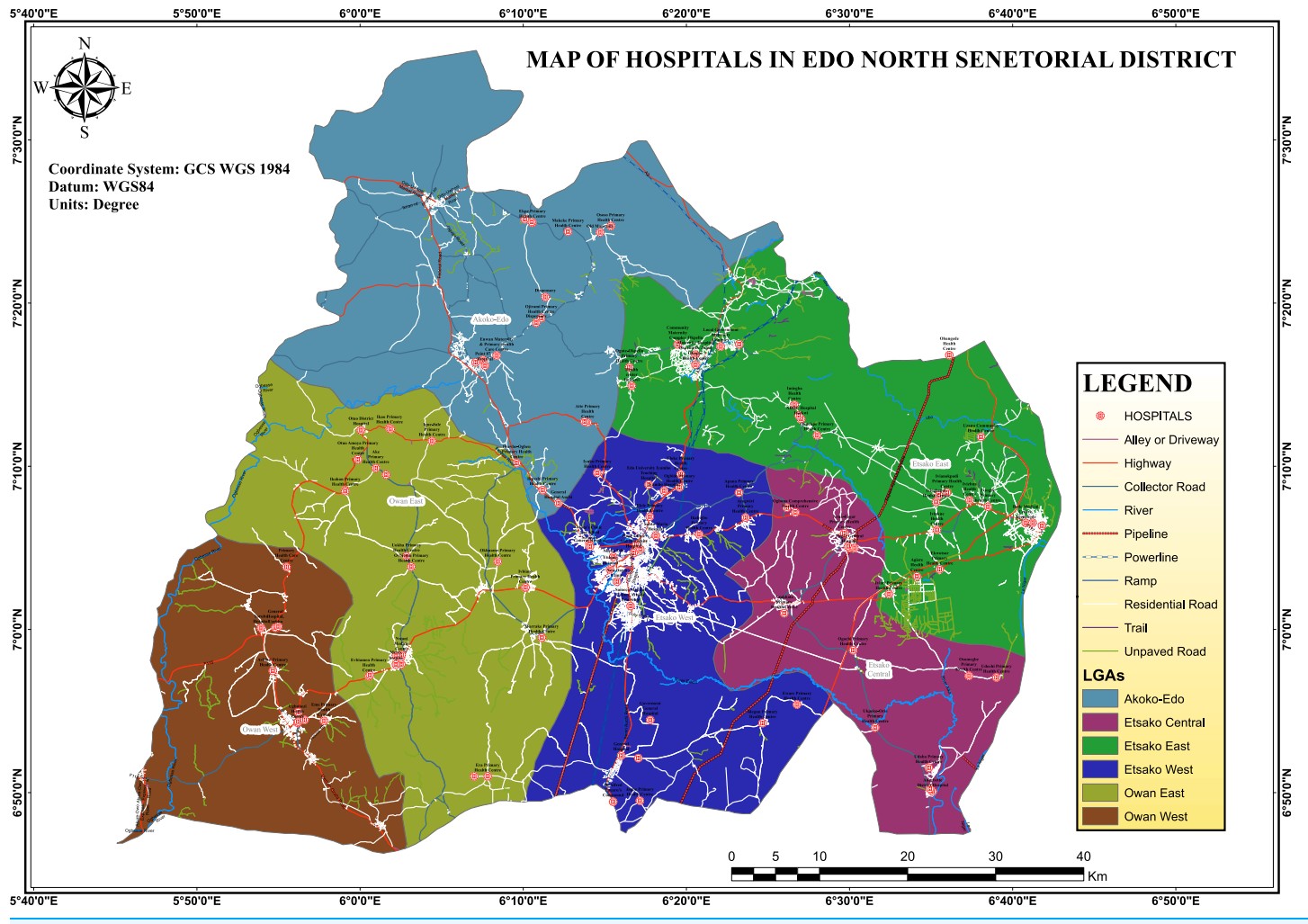

**Figure 1 Map of hospitals in Edo-North Senatorial district, Edo State, Nigeria.**

## Determination of minimum sample size

The Swinscow Formula was used to determine the minimum sample size (n) (*Charan & Biswas, 2013*).

$$n = \frac{Z^2 \times \text{Prevalence from asimilar study} \times (1 - \text{Prevalence from similar study})}{\text{Level of Precision}^2}$$

$$n = \frac{Z^2 P(1 - P)}{d^2} = 384.6$$

where *n* is the minimum sample size for this study, Z is the confidence level, d is set to be 0.05, Z is 1.96 and P is 0.5. Using the minimum sample size formula resulted in approximately 385. A total of 605 participants were sampled in six local government areas. Approximately 101 participants, including men, women and minors, were recruited from each area. Informed consent was obtained from all adult participants and from parents or guardians for minors.

### Ethical considerations and participant assent/consent

This research received ethical approval from the Health Research Ethics Committee, Irrua Specialist Teaching Hospital, Irrua, Edo State, Research Ethics Committee, with approval number: 51/23. Before collecting data, we obtained verbal assent for the children below 18 years of age through their parents or guardians. Consent forms were distributed to all participants, ensuring their agreement to donate blood samples for malaria screening. To ensure informed consent, the study objectives were explained clearly explained to each participant individually. For those with literacy limitations, the questionnaires were read aloud and questions clarified to maintain consistency in responses. Participation was entirely voluntary, and participants were informed of their right to withdraw at any time. Written informed consent, obtained through signatures or thumbprints, was collected from all participants. Confidentiality and privacy of all participants were strictly maintained throughout the research. There were no costs associated with participation in the study. In accordance with national treatment guidelines, the respondents who tested positive for malaria parasites were treated with antimalarial drugs.

### Data collection and instrumentation process

This study used a semi-structured questionnaire that was developed. A semi-structured questionnaire in English was used in this study as over 90% of the residents in Edo North communicate in English. Data collectors, consisting of final year statistics and medical laboratory science students, were trained for the field work. The questionnaire covered demographics (age, sex, occupation, marital status and educational status) and socio-economic factors (type of house, household size, type of environment, type of toilet, water sources and water storage methods) and malaria prevention practices (ownership and use of insecticide-treated nets and indoor residual spraying). The questionnaire was piloted in several locations (Auchi, Igarra, Ibillo, Ozala, Otuo, Sabongida-Ora, Afuze, Fuga, Jattu, Agenebode and Okpella) before being administered to the entire study population. For the collection of data from young children, their parents or guardians were interviewed in order to have accurate and detailed information.

### Sample collection

A digital clinical thermometer was used to take the participants' body temperatures; the thumb cleaned sterile disposable cotton wool and "lancet" is used to prick the finger in order to collect blood for microscopic analysis in the laboratory. Laboratory records of General Hospitals and Public Health Centers were used to get data on trends in hospital prevalence of malaria in Edo-North.

### Laboratory analysis

After being air-dried, the thick blood films were taken to Irrua Specialist Teaching Hospital and stained for 15 to 30 min with 5% Giemsa. They were cleaned, dried by air, and examined under an optical microscope with an X-100 objective. After examining over 100 fields with X-100 high power microscopy and finding no malaria parasite, a thick blood smear was considered negative. By comparing the number of parasites on positive slides to

200 white blood cells (WBC), they computed the parasites/µl of blood using the assumption that there are 8,000 leucocytes per microliter.

The quality control for the malaria microscopy includes regular microscope calibration and cleaning, use of high quality reagents (Giemsa stain), standardized slide preparation, the services of qualified medical laboratory scientists, cross-checking of slides by multiple microscopists, and the use of positive and negative control slides.

### Statistical analyses

The data collected was assessed, cleaned, and analyzed with the aid of R 4.3.2 software (*R Core Team, 2023*). The mean ± standard deviation was used to express the age. Malaria prevalence among participants was obtained using frequencies and percentages. The findings of the Hosmer-Lemeshow test, which was used to test the data's goodness of fit or normality, reveal a non-normal distribution. Kruskal-Wallis (H) and Mann-Whitney (W) tests were used to analyze the parasite load. The prevalence and factors influencing the parasite load were determined using the binary logistic regression model and Chi-square. Multivariate logistic regression study model variables were selected based on their significance at $p$-value < 0.05. The selection of variables based on a $p$-value of 0.05 in multivariate logistic regression aims to balance model complexity and accuracy. By focusing on significant variables, researchers aim to identify key factors influencing malaria prevalence while avoiding over fitting. Two separate multivariate logistic regression models were employed to analyze the associations between demographic factors and malaria prevalence, and between environmental and behavioral characteristics and malaria prevalence. Odd ratios; crude odd ratios (COR) and adjusted odd ratios (AOR) were computed at 95% confidence intervals. $p$-values less than 0.05 were regarded as statistically significant.

## RESULTS

### Prevalence of malaria in Edo-North

A total of 605 respondents were under investigation during the study period. The average age of the respondents was 26.40 ± 20.16 years. Table 1 shows that out of the 605 blood samples tested, 94 were microscopically malaria parasites positive, giving a total prevalence of malaria in Edo-North of 15.54%. The percentage of persons who tested positive for malaria parasite but asymptomatic to malaria infection is 13.06% while that of persons with symptomatic malaria infection who also tested positive for malaria parasite is 2.48%. The percentage of persons observed to be febrile but was not as a result of malaria parasite was 6.45%. Approximately 78.02% of the study participants were apparently free of malaria parasites. Hence, from Table 1, the malaria infection prevalence in Edo-North was statistically significant at $p$-value = 0.011.

### The relationship between malaria prevalence and demographic factors

The demographic characteristics are outlined in Table 2. Out of the 605 participants, 94 tested positive for malaria. Among these positive cases, 61.7% were male while 38.3% are female. The age distribution of the study population was as follows: 3.1% infants (9.6%

**Table 1 Prevalence of malaria in Edo-North, Edo State, Nigeria.**

| Status | Symptoms | No symptoms | Total | Odd ratio | 95% CI | *p*-value | Z statistic |
|---|---|---|---|---|---|---|---|
| Positive | 15 | 79 | 94 | 2.298 | [1.210–4.364] | 0.011 | 2.543 |
| Negative | 39 | 472 | 511 | | | | |
| Total | 54 | 551 | 605 | | | | |

prevalence), 25.5% children (44.7% prevalence), 17.9% teenagers (29.8% prevalence), and 53.6% adults (16.0% prevalence). Individuals residing in rural settlements had a significantly higher infection rate (77.7%) compared to those living in urban areas (22.3%). Akoko-Edo had the highest prevalence, followed by Etsako-West, Owan-East, Owan-West, Etsako-East, and Etsako-Central being the least. Regarding education, individuals with no formal education had the highest malaria prevalence, followed by those with primary, secondary, college, polytechnic, and university education, respectively. Pupils/students had the highest malaria prevalence among occupational groups, followed by farmers and artisans. Business owners and unemployed individuals had similar prevalence rates, while civil servants had the lowest. Marital status was associated with malaria prevalence. Unmarried individuals had the highest rates, followed by married individuals, widowed/widowers, and divorced/single parents. The malaria infection rate was notably higher among residents of mud houses than those living in modern or cement-built houses. Malaria prevalence increased with household size. Households with 10 or more people had the highest risk, while those with 1–5 people had the lowest. Malaria prevalence was highest among those using pit latrines, followed by bush toilets and water systems.

A multivariate logistic regression analysis was then performed on the significant variables, using the response variable (malaria prevalence) and the explanatory variables (socio-demographic factors). The results of the analysis included the following: Individuals with no education were approximately seven times more likely to contract malaria compared to those with higher educational levels (AOR = 6.484, 95% CI: [2.045–20.565]). Children aged 3–12 were about 11 times more likely to contract malaria (AOR = 11.324, 95% CI: [9.827–24.321]). Additionally, residents of Owan-East local government were found to be more susceptible to malaria infection (AOR = 4.617, 95% CI: [1.71–4.281]). Individuals using pit latrine had about 10 times at increased risk of malaria infection (AOR = 10.327, 95% CI: [5.473–19.485]). These variables were significantly associated with the risk of malaria.

## Environmental and behavioral characteristics of residents and malaria prevalence

Table 3 summarizes the environmental and behavioral characteristics of the survey respondents. Individuals who used indoor residual spray (IRS) had a lower risk of malaria compared to those who did not. Those who slept without long-lasting insecticidal nets (LLINs) were more susceptible to malaria infection. Residents with bushes in their surroundings exhibited a higher risk of malaria than those without. Malaria prevalence was

**Table 2 Relationship between malaria prevalence and demographic factors with total participant (N = 605) and prevalence numbers (n = 94).**

| Variables | Category | Participant N (%) | Prevalence n (%) | COR | Binary logistic (95% CI) | p-value | AOR | Multivariate logistic (95% CI) | p-value |
|---|---|---|---|---|---|---|---|---|---|
| Gender | Female | 281 (46.4) | 36 (38.3) | 1.00 | | | | | |
| | Male | 324 (53.6) | 58 (61.7) | 1.49 | [0.95–2.33] | 1.000 | 0.674 | [0.429–1.058] | 0.086 |
| Age-group (years) | Infants (1–2) | 19 (3.1) | 9 (9.6) | 18.54 | [6.56–52.400] | 0.001 | 0.472 | [0.26–1.033] | 0.843 |
| | Children (3–12) | 154 (25.5) | 42 (44.7) | 7.713 | [4.12–14.470] | 0.001 | 11.324 | [9.827–24.321] | 0.002 |
| | Teenagers (13–19) | 108 (17.9) | 28 (29.8) | 7.21 | [3.68–14.140] | 0.001 | 1.037 | [0.313–1.315] | 0.999 |
| | Adults (>19) | 324 (53.6) | 15 (16.0) | 1.00 | | | | | |
| Mean ± S.D age | 26.40 ± 20.16 | | | | | | | | |
| Location | Urban | 194 (32.1) | 21 (22.3) | 0.560 | [0.340–0.940] | 0.029 | 1.303 | [0.919–3.848] | 1.000 |
| | Rural | 411 (67.9) | 73 (77.7) | 1.00 | | | | | |
| Local government area | Akoko-Edo | 110 (18.2) | 23 (24.5) | 1.00 | | | | | |
| | Etsako-Central | 96 (15.9) | 10 (10.6) | 0.440 | [0.198–0.979] | 0.044 | 2.440 | [1.41–3.194] | 1.000 |
| | Etsako-East | 93 (15.4) | 11 (11.7) | 0.507 | [0.233–0.106] | 0.088 | 2.400 | [2.815–5.321] | 1.000 |
| | Etsako-West | 98 (16.2) | 19 (20.2) | 0.910 | [0.461–1.795] | 0.785 | 0.872 | [0.375–2.028] | 0.750 |
| | Owan-East | 103 (17.0) | 17 (18.1) | 1.323 | [0.373–1.497] | 0.041 | 4.176 | [1.710–4.281] | 0.025 |
| | Owan-West | 105 (17.4) | 14 (14.9) | 0.582 | [0.281–1.203] | 0.144 | 1.285 | [0.597–2.765] | 0.521 |
| Educational level | No education | 90 (14.9) | 28 (29.8) | 0.613 | [0.269–1.394] | 0.024 | 6.484 | [2.045–20.565] | 0.002 |
| | Primary | 63 (10.4) | 21 (22.3) | 0.679 | [0.285–1.614] | 0.380 | 0.421 | [0.360–1.421] | 0.999 |
| | Secondary | 208 (34.4) | 17 (18.1) | 0.121 | [0.052–0.283] | 0.001 | 0.311 | [0.190–1.100] | 0.999 |
| | College of education | 33 (5.5) | 14 (14.9) | 1.000 | | | | | 1.000 |
| | Polytechnic | 162 (26.8) | 9 (9.6) | 0.080 | [0.030–0.209] | 0.001 | 0.760 | [0.120–2.710] | 1.000 |
| | University | 49 (8.1) | 5 (5.3) | 0.154 | [0.049–0.489] | 0.002 | 0.741 | [0.323–1.920] | |
| Occupation | Business | 41 (6.8) | 11 (11.7) | 2.591 | [1.078–6.230] | 0.033 | 1.800 | [1.131–9.924] | 1.000 |
| | Farmer | 208 (34.4) | 22 (23.4) | 0.836 | [0.416–1.680] | 0.615 | 2.000 | [1.661–377] | 1.000 |
| | Civil servant | 36 (6.0) | 9 (9.6) | 2.356 | [0.931–5.959] | 0.070 | 1.567 | [0.604–4.063] | 0.356 |
| | Artisans | 121 (20.0) | 15 (16.0) | 1.00 | | | | | |
| | Pupils/Student | 141 (23.3) | 26 (27.7) | 1.598 | [0.803–3.179] | 0.182 | 0.505 | [0.229–1.115] | 0.091 |
| | Unemployed | 58 (9.6) | 11 (11.7) | 1.654 | [0.707–3.871] | 0.246 | 4.200 | [2.602–7.310] | 1.000 |
| Marital status | Unmarried | 361 (59.7) | 43 (45.7) | 0.116 | [0.037–0.361] | 0.001 | 3.850 | [0.579–5.069] | 1.000 |
| | Married | 204 (33.7) | 25 (26.6) | 0.120 | [0.037–0.385] | 0.001 | 2.128 | [1.143–6.29] | 1.000 |
| | Widow/Widower | 27 (4.5) | 19 (20.2) | 2.036 | [0.518–7.995] | 0.308 | 0.513 | [0.441–1.647] | 1.000 |
| | Divorced/Single Parent | 13 (2.1) | 7 (7.4) | 1.00 | | | | | |
| Religion | Christian | 397 (65.6) | | | | | | | |
| | Muslim | 194 (32.2) | | | | | | | |
| | Traditional | 9 (1.5) | | | | | | | |
| | No religion | 5 (0.8) | | | | | | | |
| House type | Modern/Cement | 452 (74.7) | 43 (45.7) | 1.00 | | | | | |
| | Mud | 153 (25.3) | 51 (54.3) | 4.756 | [3.002–7.534] | 0.001 | 5.389 | [3.590–8.900] | 1.000 |
| Household size | 1–5 | 296 (48.9) | 20 (21.3) | 0.027 | [0.013–0.055] | 0.001 | 0.100 | [0.590–6.900] | 1.000 |
| | 6–10 | 243 (40.2) | 26 (27.7) | 0.045 | [0.023–0.088] | 0.001 | 3.232 | [2.172–9.429] | 1.000 |
| | >10 | 66 (10.9) | 48 (51.1) | 1.00 | | | | | |

| Variables | Category | Participant N (%) | Prevalence n (%) | COR | Binary logistic (95% CI) | p-value | AOR | Multivariate logistic (95% CI) | p-value |
|---|---|---|---|---|---|---|---|---|---|
| Toilet type | Water system | 366 (60.5) | 14 (14.9) | 0.055 | [0.027–0.110] | 0.001 | 0.800 | [0.423–1.913] | 0.000 |
| | Pit latrine | 158 (26.1) | 46 (48.9) | 0.568 | [0.325–0.993] | 0.047 | 10.327 | [5.473–19.485] | 0.001 |
| | Bush toilet | 81 (13.4) | 34 (36.2) | 1.00 | | | | | |

**Table 3 Environmental and behavioral characteristics of residents and malaria prevalence in Edo North, Nigeria with total participant (N = 605) and prevalence numbers (n = 94).**

| Variable | Category | Participant N (%) | Prevalence n (%) | COR | Binary logistic (95% CI) | p-value | AOR | Multivariate logistic (95% CI) | p-value |
|---|---|---|---|---|---|---|---|---|---|
| Do you use indoor residual spray (IRS)? | Yes | 417 (68.9) | 63 (67.0) | 0.901 | [0.564–1.441] | 0.664 | 1.620 | [1.506–1.819] | 1.000 |
| | No | 188 (31.1) | 31 (33.0) | 1.000 | | | | | |
| Do you utilize LLINs | Yes | 142 (23.47) | 87 (89.362) | 1.000 | | | | | |
| | No | 463 (76.529) | 7 (7.447) | 0.403 | [0.257–0.638] | 0.001 | 0.327 | [0.319–0.742] | 0.003 |
| Presence of bushes in surroundings | Yes | 394 (65.1) | 76 (80.9) | 1.130 | [1.487–4.415] | 0.001 | 2.563 | [1.947–5.322] | 0.025 |
| | No | 211 (34.9) | 18 (19.1) | 1.000 | | | | | |
| Presence of stream and rivers around home | Yes | 54 (8.9) | 37 (39.4) | 18.863 | [9.982–35.643] | 0.001 | 1.326 | [0.57–2.392] | 0.002 |
| | No | 551 (91.1) | 57 (60.6) | 1.000 | | | | | |
| Use of window-nets | Yes | 478 (79.0) | 60 (63.8) | 1.000 | | | | | |
| | No | 127 (21.0) | 34 (36.2) | 0.393 | [0.244–0.633] | 0.001 | 0.140 | [0.032–0.852] | 0.997 |
| Types of ceiling | PVC | 174 (28.8) | 19 (20.2) | 0.334 | [0.186–0.602] | 0.001 | 0.850 | [0.301–2.397] | 0.758 |
| | POP | 53 (8.8) | 5 (5.3) | 0.284 | [0.106–0.760] | 0.012 | 1.266 | [0.542–2.957] | 0.586 |
| | Asbestos | 164 (27.1) | 44 (46.8) | 1.000 | | | | | |
| | Plywood | 67 (11.1) | 9 (9.6) | 0.423 | [0.194–0.926] | 0.031 | 0.925 | [0.021–40.551] | 0.968 |
| | No ceiling | 147 (24.3) | 17 (18.1) | 0.357 | [0.193–0.658] | 0.001 | 0.362 | [0.030–4.422] | 0.426 |
| Source of water | Bore hole | 123 (20.3) | 14 (14.9) | 1.000 | | | | | |
| | River/Stream | 294 (48.6) | 30 (31.9) | 0.885 | [0.452–1.733] | 0.721 | 2.251 | [1.234–4.108] | 0.008 |
| | Rain | 108 (17.9) | 22 (23.4) | 1.992 | [0.962–4.122] | 0.063 | 4.738 | [2.614–8.588] | 0.000 |
| | Dam/Tap | 80 (13.2) | 28 (29.8) | 4.192 | [2.937–8.627] | 0.001 | 1.130 | [0.577–2.214] | 0.721 |
| Types of water storage containers | Closed | 172 (28.4) | 23 (24.5) | 0.154 | [0.046–0.520] | 0.003 | 10.016 | [2.566–14.764] | 1.000 |
| | Open | 421 (69.6) | 65 (69.1) | 0.183 | [0.057–0.564] | 0.004 | 0.508 | [0.004–0.943] | 0.045 |
| | Both | 12 (2.0) | 6 (6.4) | 1.000 | | | | | |

higher among individuals living near rivers or streams compared to those with other water sources. Additionally, storing water in open containers was associated with a higher risk of malaria compared to other storage methods.

After adjusting for possible confounding odds (AOR) of environmental and behavioural characteristics of the residents and malaria prevalence, the use of long-lasting insecticidal nets (LLINs) is associated with a lower risk of malaria infection. Residents with bushes in

their surroundings had a three times at increased risk of malaria infection (AOR = 2.563, 95% CI: [1.947–5.322]). Additionally, those living near streams or rivers were approximately three times more susceptible to malaria (AOR = 2.563, 95% CI: [1.057–2.392]). Individuals who relied on rivers/streams or rainwater as their primary water source were significantly more likely to contract malaria. Compared to those using other sources, their risk was approximately three times higher for river/stream water and four times higher for rain water (AOR = 2.251, 95% CI: [1.234–4.108] and AOR = 4.7388, 95% CI: [2.614–8.588]) respectively. Storing water in open containers was also associated with an increased risk of malaria infection (AOR = 10.016, 95% CI: [2.566–14.764]). Malaria infection was significantly associated with these five environmental and behavioural characteristics of the residents' variables.

## Malaria prevalence and management factors in human populations

From Table 4, the analysis showed that 87.2% of participants had experienced malaria in the past. Of these, 71.3% had received treatment in the past 1–5 months, 20.2% in the past 6–12 months, and 8.5% in the past year. The most common source of treatment was health centers (40.4%), followed by medical/dispensaries centers (22.3%), pharmacies/chemists (20.2%) and traditional/herbal homes (17.0%).

The multivariable logistic regression analysis revealed that individuals who had contracted malaria in the past were significantly more susceptible to future malaria infections, with an odds ratio of 4.839 and a 95% confidence interval of 2.187 to 10.708. Individuals with a history of malaria who received treatment from pharmacies or chemist shops were approximately six times more likely to have recurrent malaria infections (95% CI: [3.232–17.867]). Conversely, those who sought treatment from traditional or herbal healers were less likely to experience recurrent malaria (95% CI: [0.237–0.984]).

## Geometric mean malaria parasite density in relation to demographic factors

From Table 5, the average geometric mean parasite density (GMPD) for malaria was approximately 5,146 parasites/μL of blood. At $p = 0.002$, the GMPD of men (7,765.48 parasites/μL) was higher than that of women (4,786.16 parasites/μL). Similarly, farmers had a higher GMPD of malaria (6,654.43/μL) than people in other occupations or professions ($p < 0.001$). When marital status was taken into consideration, single persons had a higher GMPD (6,542.32 parasites/μL) than other groups at the significance level of 5%. For age group, teenager had higher GDPM (5,929.00 sites/μL). Those who have live for 11–15 years had higher GDPM (4,126.46 sites/μL) while those who have lived in their locations for over 15 years had lower malaria parasite density load. For location of participants, 8,876.03 parasites/μL, residents of Owan-East had the highest geometric mean malaria parasite density, followed by Owan-West (656.65 parasites/μL), Etsako-Central (6,545.23 parasites/μL), Etsako-East (6,087.65 parasites/μL), Etsako-West (5,876.31 parasites/μL) and Akoko Edo (5,345.87 parasites/μL).

**Table 4 Malaria prevalence and management factors in human populations in Edo North, Edo State, Nigeria with total participant ($N = 605$) and prevalence numbers ($n = 94$).**

| Variable | Category | Participant N (%) | Prevalence n (%) | Binary logistic COR | (95% CI) | p-value | Multivariate logistic AOR | (95% CI) | p-value |
|---|---|---|---|---|---|---|---|---|---|
| Have you had malaria before? | Yes | 578 (95.5) | 82 (87.2) | 1.504 | [0.647–2.453] | 0.229 | 4.839 | [2.187–10.708] | 0.000 |
| | No | 27 (4.5) | 12 (12.8) | 1.000 | | | | | |
| If yes, when last were you sick of malaria? | 1–5 months ago | 243 (40.2) | 67 (71.3) | 3.000 | [1.412–5.230] | 0.001 | 1.125 | [0.645–3.387] | 0.242 |
| | 6–12 months ago | 324 (53.6) | 19 (20.2) | 2.163 | [0.635–10.651] | 0.001 | 1.303 | [0.764–3.765] | 0.250 |
| | Over 1 year ago | 38 (6.3) | 8 (8.5) | 1.000 | | | | | |
| Where did you take treatment? | Medical hospital | 91 (15.0) | 21 (22.3) | 1.000 | | | | | |
| | Health center | 185 (30.6) | 38 (40.4) | 1.167 | [0.342–17.243] | 0.632 | 3.244 | [1.040–12.277] | 0.514 |
| | Pharmacy/Chemist shop | 228 (37.7) | 19 (20.2) | 1.505 | [0.117–7.345] | 0.001 | 6.325 | [3.232–17.867] | 0.001 |
| | Tradomedical/ Herbs | 101 (16.7) | 16 (17.0) | 3.500 | [0.134–13.233] | 0.001 | 0.483 | [0.237–0.984] | 0.045 |
| What drugs did you use in treating malaria? | Athermeter family | 304 (50.2) | 53 (56.4) | 1.000 | | | | | |
| | Chloroquine/ Quinine | 45 (7.4) | 11 (11.7) | 2.325 | [0.832–5.648] | 0.021 | 2.380 | [1.432–11.431] | 0.005 |
| | Use of herbs (Agbo) | 101 (16.7) | 12 (12.8) | 4.903 | [1.745–12.432] | 0.005 | 0.974 | [0.448–2.121] | 0.948 |
| | Don't know | 155 (25.6) | 18 (19.1) | 0.662 | [0.246–1.603] | 0.768 | 2.400 | [0.967–5.953] | 0.059 |

## Assessment of long-lasting insecticidal net ownership and utilization in Edo-North

From Fig. 2, LLIN coverage in Edo North is 76.53%, with approximately 463 of the 605 respondents reporting having LLINs at home. From Fig. 3, the majority (61.98%) of people with LLINs rarely sleep under them on a daily basis, while 27.50% sleep under them frequently and 10.41% do not use them. Figure 4 shows that the majority of nets (43.54%) were distributed free of charge by the government, followed by those purchased by individuals (42.62%). A smaller proportion (14.84%) were obtained by other means.

## Malaria prevalence patterns in Edo-North between 2000 and 2023

From Fig. 5, records from health facilities and hospitals show that between 2020 and 2023, the prevalence of malaria among residents of Edo North is on a downward trend.

## DISCUSSION

Malaria affects the lives of almost many people living in sub-Saharan African countries, including Nigeria. Despite continuous control and preventive strategies in place, malaria remains a major public health problem in Nigeria. The overall prevalence of malaria in this study was 18.58%. On demographic factors, gender, age, location, occupation, and marital status were all associated with malaria prevalence. Males (61.7%) were more likely to have malaria than females (38.3%). This is consistent with previous studies in Gabon and Kampala International University Teaching Hospital Bushenyi Western Uganda (*Mawili-*

**Table 5 Geometric mean malaria parasite density in relation to demographic factors in Edo North, Edo State, Nigeria.**

| Characteristics<br>Overall GMPD | Category | Number examined | GMPD (Parasites/µL)<br>5,146 | Statistics | p-value |
|---|---|---|---|---|---|
| Sex | Female | 281 | 5,786.66 | W = 55.0 | 0.002* |
| | Male | 324 | 7,765.48 | | |
| Occupation | Business | 41 | 7,237.45 | H = 2.00 | 0.001* |
| | Farmer | 208 | 6,654.43 | | |
| | Civil servant | 36 | 4,765.76 | | |
| | Artisans | 121 | 5,434.21 | | |
| | Student/Pupils | 141 | 5,436.37 | | |
| | Unemployed | 58 | 3,237.65 | | |
| Marital status | Single | 361 | 6,542.32 | H = 12.45 | 0.001* |
| | Married | 204 | 5,432.43 | | |
| | Widow/Widower | 27 | 2,242.65 | | |
| | Divorced/Single parent | 13 | 1,221.32 | | |
| Age group | Infants (1–2) | 19 | 5,929.00 | H = 14.32 | 0.001* |
| | Children (3–12) | 154 | 2,047.00 | | |
| | Teenagers (13–19) | 108 | 2,745.00 | | |
| | Adults (>19) | 324 | 806.00 | | |
| Duration of stay | 0–5 years | 110 | 3,857.06 | H = 9.76 | 0.001* |
| | 6–10 years | 96 | 1,089.14 | | |
| | 11–15 years | 93 | 4,126.46 | | |
| | 16–20 years | 98 | 812.11 | | |
| | 21–30 years | 103 | 924.21 | | |
| | >30 years | 105 | 781.28 | | |
| LGA resident | Akoko-Edo | 110 | 5,345.87 | H = 16.08 | 0.025* |
| | Etsako-Central | 96 | 6,545.23 | | |
| | Etsako-East | 93 | 6,087.65 | | |
| | Etsako-West | 98 | 5,876.31 | | |
| | Owan-East | 103 | 8,876.03 | | |
| | Owan-West | 105 | 656.65 | | |

**Notes:**
* Significant association at $p < 0.05$.
W = Mann-Whitney Statistic and H = Kruskal-Wallis Statistic.

*Mboumba et al., 2013*; *Ogah et al., 2013*). Compared with other age groups, children had a significantly higher prevalence of malaria (44.7%), which is in line with (*Yutura et al., 2024*; *Singh et al., 2014*; *World Health Organization (WHO), 2009*). The study found that rural residents were more likely to be affected than urban residents, with prevalence rates of 67.9% and 32.1%, respectively. The prevalence varied between the study locations, with the highest prevalence in Akoko-Edo (24.5%), followed by Etsako-West (20.2%), Owan-East (18.1%), Owan-West (14.9%), Etsako-East (11.6%) and Etsako-Central (10.6%).

The findings from the study provide valuable insights into the environmental and behavioral factors associated with malaria prevalence in the study population. The use of
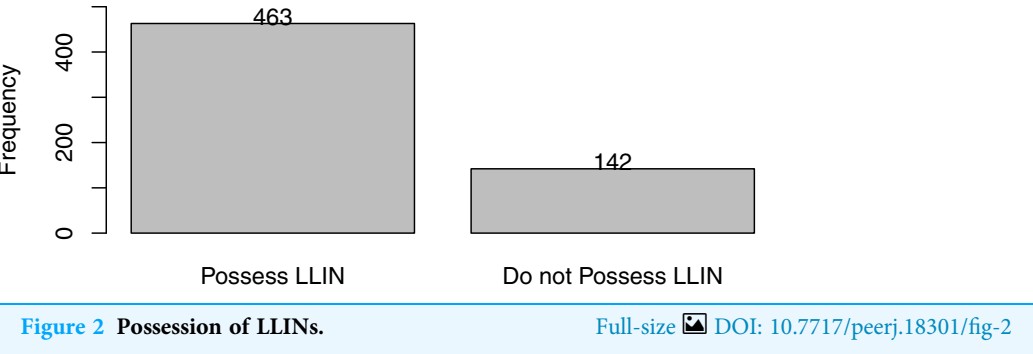

**Figure 2  Possession of LLINs.**               

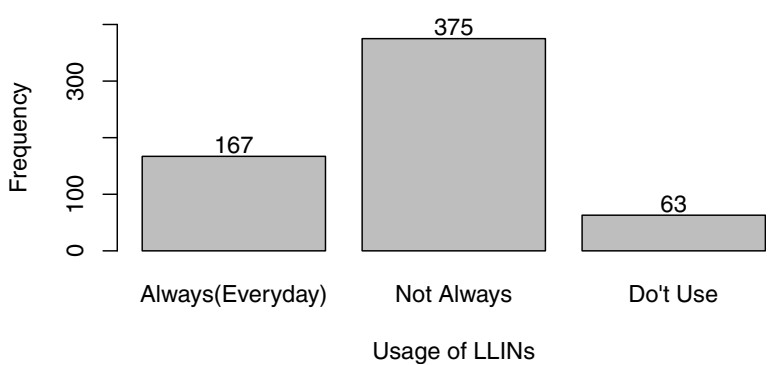

**Figure 3  Frequency of use of LLINs.**         

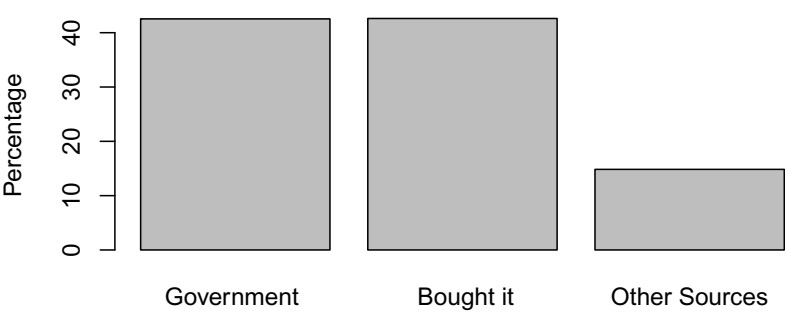

**Figure 4  Source of acquisition of LLINs.**    

indoor residual sprays (IRS) and long-lasting insecticidal nets (LLINs) were found to be protective against malaria infection, while exposure to mosquito breeding sites, such as bushes and proximity to water bodies, increased the risk of disease. The multivariable logistic regression analysis confirmed the significant association between several environmental and behavioral factors and malaria prevalence. Individuals who did not sleep under LLINs were nearly three times more likely to contract malaria, highlighting the

8000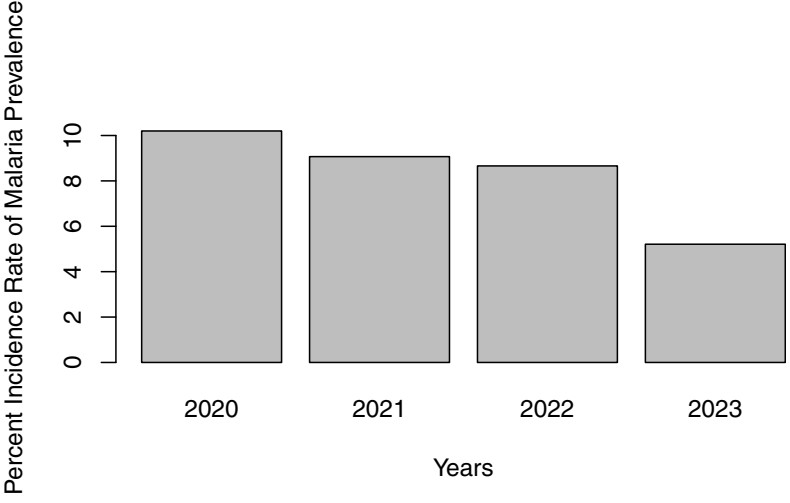

**Figure 5 Malaria incidence trend from medical records, Edo-North, 2000–2023.**

importance of LLINs as a crucial tool for malaria prevention. Residents with bushes in their surroundings had a 50% increased risk of malaria infection, emphasizing the need for environmental management strategies to reduce mosquito breeding sites. Proximity to streams or rivers was a strong predictor of malaria infection, suggesting that these water bodies are important breeding grounds for mosquitoes. Storing water in open containers was also associated with an increased risk of malaria, as these containers can serve as breeding sites for mosquito larvae. The study provide valuable insights into the prevalence, treatment seeking behaviour and risk factors for malaria infection in the study population. The prevalence of malaria was high, with 87.2% of participants reporting a history of the disease. The multivariate logistic regression analysis revealed that children, residents of Owan-East, individuals with no formal education, those using pit latrines, those not using LLINs, and those living near bushes, streams, or rivers were at a higher risk of malaria infection. Furthermore, the treatment and medication selected played a significant role in determining the risk of recurrent malaria. That is, individuals treated at pharmacies/chemist shops and those who were treated with chloroquine/quinine medication were associated with a significantly higher risk of malaria.

The findings also provides valuable insights into the factors associated with malaria parasite density in the study population. The average geometric mean parasite density (GMPD) of 5,146 parasites/μL of blood aligns with previous studies conducted in malaria-endemic regions (*Singh et al., 2014*). However, the significant differences observed among various demographic and socio-demographic factors highlight the interplay of factors influencing malaria transmission. The higher GMPD in men compared to women is consistent with findings from other studies (*Riggs et al., 2020*). This could be attributed to several factors, including differences in occupational exposure, behavior, and biological factors. Men may engage in more outdoor activities, increasing their risk of exposure to mosquito bites. Farmers exhibited a significantly higher GMPD than individuals in other occupations. This is likely due to their frequent exposure to agricultural fields, which are

8000

often breeding grounds for mosquitoes (*Oladepo et al., 2010*). Single individuals were found to have a higher GMPD than those in other marital statuses. This could be attributed to differences in social interactions, housing conditions, and access to healthcare. Teenagers and individuals aged 11–15 years had higher GMPDs compared to older age groups. This suggests that younger individuals may be more susceptible to malaria infection due to factors such as immature immune systems or differences in behavior (*Singh et al., 2014*). However, the lower GMPD in individuals who have lived in their locations for over 15 years could be attributed to acquired immunity or changes in the local mosquito population over time. Residents of Owan-East had the highest GMPD, followed by Owan-West, Etsako-Central, Etsako-East, Etsako-West, and Akoko Edo. These variations in GMPD across different locations are likely influenced by factors such as ecological conditions, mosquito density, and access to healthcare services. Regions with higher mosquito populations and lower levels of healthcare access may have higher rates of malaria transmission.

For the prevalence trend, a downward trend was observed. The observed decline in malaria prevalence between 2020 and 2023 is encouraging and suggests that ongoing malaria control efforts are having a positive impact.

## LIMITATIONS AND FUTURE RESEARCH

This study was cross-sectional, limiting the ability to establish causality. Longitudinal studies are needed to further explore the temporal dynamics of malaria transmission in the region. Additionally, incorporating molecular diagnostic techniques to identify specific malaria parasite species could provide valuable insights into the epidemiology of malaria in Edo-North.

## CONCLUSIONS AND RECOMMENDATIONS

The study revealed a significant malaria burden in Edo North, Nigeria, with factors such as age, occupation, housing and environmental factors significantly influencing the risk of infection. The observed decline in malaria prevalence from medical records from 2020 to 2023 is encouraging, but requires sustained control efforts.

Based on the findings of this study, the following recommendations can be made: In order to strengthen vector control, there should be increase of LLINs distribution and ensure consistent usage, promote IRS programs, and improve sanitation and environmental management practices. Improve access to healthcare services, especially in rural areas, and ensure availability of appropriate antimalarial drugs. There should be a continuous monitoring of malaria prevalence and identifying emerging trends to inform targeted interventions. Promoting community awareness and participation in malaria prevention and control activities. Future research should focus on longitudinal studies and molecular diagnostics to improve our understanding of malaria transmission dynamics in the region.

By implementing these recommendations, Edo-North can further reduce the burden of malaria and improve the health and well-being of its population.

## ACKNOWLEDGEMENTS

The authors are grateful to the health centre workers in the study areas, the correspondents, the final year medical laboratory science students at Ambrose Alli University who participated, and other medical officers for their cooperation and sharing of information during this study.

### Funding

The authors received no funding for this work.

### Competing Interests

The authors declare that they have no competing interests.

### Author Contributions

- Joseph Odunayo Braimah conceived and designed the experiments, performed the experiments, analyzed the data, authored or reviewed drafts of the article, and approved the final draft.
- Nnamdi Edike performed the experiments, analyzed the data, prepared figures and/or tables, and approved the final draft.
- Augustine Ijeameran Okhaiomoje conceived and designed the experiments, performed the experiments, authored or reviewed drafts of the article, and approved the final draft.
- Fabio Mathias Correa performed the experiments, analyzed the data, prepared figures and/or tables, authored or reviewed drafts of the article, and approved the final draft.

### Human Ethics

The following information was supplied relating to ethical approvals (*i.e.*, approving body and any reference numbers):

Health Research Ethics Committee, ISTH (Ref: 51/23) granted ethical approval to carry out the study, using its microbiology laboratory for Malaria rapid diagnostic tests (RDTs).

### Data Availability

The raw data is available in the Supplemental Files.

### Supplemental Information

Supplemental information for this article can be found online at http://dx.doi.org/10.7717/peerj.18301#supplemental-information.

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
