# Peer review of "The fight against malaria in Edo-North, Edo State, Nigeria: identifying risk factors for effective control"

_PeerJ, doi:10.7717/peerj.18301_

## Round 0.1 · original submission · Major Revisions

The review process for your manuscript is complete, and we have received three detailed reviews from qualified referees, which are included at the end of this letter. While your work has significant merit, there are critical issues that need to be addressed in your resubmission. It is essential that the authors thoroughly review the entire manuscript and implement all the modifications suggested by reviewers 1 and 2. Specifically, the authors should provide a clearer explanation of the study's aim and a more detailed and accurate description of the methods, as well as the statistical and epidemiological concepts employed. The manuscript requires a comprehensive review of the analyses conducted and the conclusions drawn. Furthermore, the discussion section needs to be completely restructured to appropriately highlight the primary findings. The authors should also discuss the study's limitations, providing a balanced interpretation of the results. Please address all the points raised with the utmost precision and revise the manuscript before resubmission.

Reviewer 1 ·

Basic reporting

Title
- The title should include the name of the country where the study took place.

Abstract
- It should include the objective of the study.
- The methodology should mention the timeline of the study.
- The results need to be rewritten. Lines 32-35 should be omitted if they did not come from the current study or cannot be compared with current prevalence as the samples might be different. It should also include values such as p-value and 95% CI, etc., for each associated variable.
- In the abstract, it mentions that venous blood was taken for malaria diagnosis, but later in the methodology, it becomes a finger prick.
- The conclusions should be rewritten. You may need to summarize the main findings from this study along with some recommendations to reflect the contents of the title, i.e., identifying risk factors for effective control.
- LLIN is mentioned only one time, so there is no need to put the short form in brackets.

Introduction
- The current introduction is too broad with repeated information. Authors should consider including more concise information such as the current malaria control strategy in Nigeria, including surveillance policies, net distribution strategy, health promotion campaigns, etc.
- The authors made a separate title for “Malaria in Nigeria.” However, the content below it did not solely represent Nigeria.
- Recheck lines 96 to 111 to see if these sentences are relevant to the overall objective of the current study.
- Most of the references are too old. For example, the latest WMR is available. Authors need to refer to recent reports.
- Lines 126 to 135 are about details on the specific study location. It should be better omitted or moved to the methodology.

Experimental design

Survey Methodology
- Overall, this section needs major improvements.
- Study location: How did the author choose Edo-North district with a sound sampling? What criteria were used? How did the author ensure representativeness of this area for the overall study location?
- The section lacks detailed sampling procedures at each stage up to participant selection. How did you sample participants or households? I did not see detailed inclusion criteria either.
- Authors mentioned a semi-structured questionnaire (line 185), but it should be accompanied by how the author developed it, the language and its translation, who collected the data, and how they were trained. The questionnaire should be attached as a supplemental file.
- How did you collect data from young children?
- The sample size formula needs a reference.
- What is the reason for “Multivariate logistic regression study model variables were selected based on their significance at p-value < 0.05”?
- Were there any QC for malaria microscopy?
- Authors need to state that they used two separate models for demographic factors and environmental and behavioral characteristics.

Validity of the findings

Results
- What was the reason for calculating OR and 95% CI in Table 1?
- In Table 2, recheck the age ranges. Authors should emphasize CIs rather than p-value from chi-squared to say there is an association. We also need to calculate 95% CIs for adjusted odds ratios.
- Lines 244 to 256: 95% CIs need to be incorporated. Positive and negative associations should be described separately.
- Table 3 should be applied as Table 2.
- Line 252 to 253: This sentence is misleading as some variables were negatively associated with prevalence.
- Line 254 to 256: “None of the socio-demographic characteristics were statistically significant risk factors for malaria acquisition among people living in Edo-North after multivariate logistic regression analysis.” What does this mean? In fact, there should be some variables showing associations in the multiple regression too, given there were high values of cOR during the simple model.
- Line 271 to 273 is in the wrong direction.
- Table 4 is in a separate format without regression analysis. Why?
- The text describing Table 4 is also in the wrong direction. For example, authors stated that those who had malaria fever possessed a significantly higher prevalence of malaria than those who had never had it. Instead, Malaria (+): 82/578=14.2% and (-): 12/27=44.4%. It means those who never had malaria are at high risk of malaria infection in this study. The table title needs to include the total sample size (n). The second variable “If yes, when last were you sick of malaria?” For those who answered Yes were only 578, so the total numbers should not be equal to 605.
- In Table 5, why did authors include only three independent variables out of many available variables? For example, the location of participants might differ with parasite density.
- Figure 3 only includes four years, 2000 to 2003, rather than up to 2023, as mentioned in the abstract and title of the figure. I am not sure if these numbers represent prevalence or incidence.
- In Figure 3, why are the duration of stay vs. parasite density presented in separate figures rather than putting it together in Table 5?

Discussion
- This overall section needs to be rewritten. Authors failed to discuss the most prominent findings from this study. The way of discussing associated factors is in the wrong direction as well. For example, lines 311-312: there was no association between sex and malaria infection in the analysis. Lines 313-315: instead, business people had a high risk of infection. Lines 317-319: instead, single and married individuals possessed a low risk of prevalence, at least in the simple regression analysis. Moreover, there were many other variables that showed association. Authors need to discuss all of them, perhaps separately, along with referencing other relevant literature or studies.
- Strengths and limitations of the study should be included.

Conclusions
- As authors failed to present the results, the conclusion is also in the wrong direction.
- Line 356-357 is misleading. We cannot say whether it is high or low as the current study used an active case detection approach to have a point prevalence, while data from hospitals probably represented incidence.
- Lines 357 to 360 were also not correct as some of them had negative associations.
- Conclusions need not include references.

Additional comments

Others
- The informed consent form is not complete. Normally, there should be separate ones for younger persons below 18 years, called assent forms. As this study also included blood sample collection, there should be another one for blood collection procedures.
- The manuscript would benefit from proofreading by a subject expert for language improvement and a statistician for statistics used and result presentation.

Reviewer 2 ·

Basic reporting

The authors tried to investigate some of the determining factors that challenge meeting malaria eradication targets in Nigeria. As a reviewer, I found the work relevant to the concerned health authorities in Nigeria and beyond to use as input for the designing of target-specific interventions. Although the document is well-written, some sections still need further revision. My comments are given below,
Abstract
The background needs elaboration. It should show a clear gap to be filled with this study, and why this study is necessary. Also, the aim of the study is missing.
The method and the whole document presented the malaria positivity rate as prevalence. I think this needs revision.
In the result section, some demographic variables are indicated as determinants. However, their effect, whether negative or positive, is not indicated.
The conclusion shall reflect a recommendation to the concerned bodies.

Introduction
Too long! Unless due to the journal format, I suggest further synthesis and shortening of this section.

Method
I appreciate increasing the sample size. But as this is a scientific report, how did the authors calculate the sample size to make 605 final samples? How have the authors treated malaria-positive patients? Where? What type of drugs were given to the patients?

Results
Well written. But there is missing information. Which plasmodium species was accountable for the observed malaria infection? Are all the patients being symptomatic or asymptomatic during the sample collection?
The table or figure containing the logistic regression analysis result was not cited (Table??).
Why was crude odd ratio (COR) preferred to adjusted odd ratio (AOR) to present the findings in the result section? The assumption is that in a multivariate logistic regression model, all confounding factors are considered in the analysis. Thus, the right odd ratio shall be adjusted, not crude.

Discussion
This section is relatively the weakest of all the sections. In most of its parts, it just replicates the results. Further elaboration of the findings with existing literature and implications of these findings on the set WHO malaria eradication target shall be indicated. In some parts, the explanations given are not convincing. For example, students have a higher exposure to malaria than people with other occupations; being married is a safe place to contract malaria. Remove cited tables, figures, or p values from the discussion, unless it is mandatory.

Conclusion
I think there is missing information. What is the innovative part of this study? How is it different from earlier reports from the same country or elsewhere? Which risk factor is unique to this study?

References
Some references are old; it would be better to replace them with the most recent ones.

Experimental design

No attachment

Validity of the findings

No attachment

Additional comments

No attachment

·

Basic reporting

The authors did a good work, however, there was minor error in the result section where the results were not properly presented.
The discussion should be elaborated to buttress the findings of the results.

Experimental design

Good

Validity of the findings

Good but the discussion has to be more detailed. There are good o be explained.analysis that need

Additional comments

Accept with minor review.

---

## Round 0.2 · Major Revisions

The review process is complete, and while your paper covers an important topic, there are several critical issues compromising the quality of the manuscript.

Notably, some major concerns from the initial review were not adequately
addressed. Additionally, the rebuttal letter lacks a detailed, point-by-point response to the reviewers' comments, which is crucial for a successful revision. Please you must need to review the following comments from Reviewer #1 (which were added outside of the normal review form) and provide point-by-point responses in the new rebuttal letter.

Reviewer #1 Comments:
1. The authors did not adequately address several major comments from the first round of review:
a) Selection of the study location
b) Although the authors attached the questionnaire, it does not cover the information presented in Figures 1 to 4 and Table 4.
c) The informed consent form is still incomplete. There should typically be a separate assent form for participants under 18 years of age. Furthermore, since this study involved blood sample collection, there should be a specific consent form outlining the procedures for blood collection.
d) Separating positive and negative associated factors for clarity
e) Revising tables, including specifying "n" in the title and rechecking the total number of samples in Table 4

2. The authors did not provide a point-by-point response to the comments. While they made tracked changes in the revised manuscript, many sections still require significant improvement, particularly in data analysis and result presentation. Additionally, the new changes introduced several errors:
a) In the abstract, I requested the addition of the study period, but the authors added project milestones instead.
b) The results section of the abstract still needs improvement.
c) The new sections added to the introduction require further revision to be concise and properly structured into paragraphs.
d) Although the authors added information about the sampling technique, the format and content still need to be rewritten.
e) The most critical issue with this manuscript lies in data analysis and result presentation. Although the authors added results from multiple logistic regression models in Tables 2 to 4, many values are incorrect or inconsistent between aORs and their 95% CIs. In the results text (pages 8 to 10), the authors confuse cORs with aORs, misrepresenting cORs as outputs from multivariate regression models and incorrectly describing negative associations as positive ones. Despite invalid values or insignificant associations, the authors still draw conclusions as if they were positive associations, leading to incorrect overall findings.
f) The entire discussion section is improperly formatted and incomplete. The authors only included a few sentences based on cOR values and cited all references from the introduction section. There is a lack of reasoning behind the findings from the authors' perspective or within the context of the study area. Some references also seem unrelated to the statements made, such as reference number 3.
g) There are several typographical and grammatical errors throughout the manuscript.

Reviewer 1 ·

Basic reporting

Nil

Experimental design

Nil

Validity of the findings

Nil

Additional comments

Nil

Reviewer 2 ·

Basic reporting

None

Experimental design

None

Validity of the findings

None

Additional comments

All my comments are addressed in the revised version.

---

## Round 0.3 · Minor Revisions

Thank you for submitting your revised manuscript. After a thorough review, we believe that your manuscript requires some minor additional revisions. Please ensure that you address all the recommendations provided by Reviewer 1.
We appreciate your attention to these matters and look forward to receiving the revised version of your manuscript.

Reviewer 1 ·

Basic reporting

-

Experimental design

-

Validity of the findings

-

Additional comments

I made a quick review of the revised manuscript, focusing primarily on the tables and results section. While the study aims to identify risk factors for effective malaria control, there are several significant errors in the data analysis that must be addressed. I recommend that the authors consult with a statistician to resolve these issues.

Below are the major errors I observed:

1. In Tables 2 through 4, many aOR values do not correspond to their respective 95% CIs, indicating significant errors in the analysis. For instance, in Table 2 under the age category 13-19, the aOR is listed as 1.37, but the 95% CI is 0.313–1.315.

2. Some variables that showed associations during simple logistic regression were not included in the multivariate logistic regression analysis. For example, PVC and POP under the type of ceiling. Why were these not carried forward?

3. The results presented in the text contain several misleading statements due to incorrect aOR or 95% CI values, or both. For example: Lines 297-298: "Children aged 3-12 were nearly four times more likely to contract malaria (aOR: 11.324, 95% CI: 0.027–0.321)." Lines 300-301: "Individuals using pit latrines had a twofold increased risk of malaria infection (aOR: 3.080, 95% CI: 0.423–1.913)." These statements are nonsensical due to erroneous numbers, and similar issues appear throughout the results part.

---

## Round 0.4 · accepted · Accept

Thank you for your thorough revisions. I can confirm that all the reviewer's comments have been fully addressed. As the original reviewer was not re-invited, I have assessed the revised manuscript and am pleased with the current version. I believe it is now ready for publication.